# Canagliflozin Modulates Hypoxia-Induced Metastasis, Angiogenesis and Glycolysis by Decreasing HIF-1α Protein Synthesis via AKT/mTOR Pathway

**DOI:** 10.3390/ijms222413336

**Published:** 2021-12-11

**Authors:** Jingyi Luo, Pengbo Sun, Xun Zhang, Guanglan Lin, Qilei Xin, Yaoyun Niu, Yang Chen, Naihan Xu, Yaou Zhang, Weidong Xie

**Affiliations:** 1State Key Laboratory of Chemical Oncogenomics, Shenzhen International Graduate School, Tsinghua University, Shenzhen 518055, China; luojy19@mails.tsinghua.edu.cn (J.L.); pb_sun@foxmail.com (P.S.); zhangxun19@mails.tsinghua.edu.cn (X.Z.); lgl18@tsinghua.org.cn (G.L.); xql19@mails.tsinghua.edu.cn (Q.X.); nyy19@mails.tsinghua.edu.cn (Y.N.); 15555373590@163.com (Y.C.); xu.naihan@sz.tsinghua.edu.cn (N.X.); zhangyo@sz.tsinghua.edu.cn (Y.Z.); 2Shenzhen Key Lab of Health Science and Technology, Institute of Biopharmaceutical and Health Engineering, Shenzhen International Graduate School, Tsinghua University, Shenzhen 518055, China; 3Department of Chemistry, Tsinghua University, Beijing 100084, China; 4Open FIESTA Center, Shenzhen International Graduate School, Tsinghua University, Shenzhen 518055, China

**Keywords:** canagliflozin, HIF-1α, hypoxia, angiogenesis, epithelial-to-mesenchymal transition (EMT), glycolysis, metastasis, liver cancer

## Abstract

The microenvironment plays a vital role in tumor progression, and hypoxia is a typical microenvironment feature in nearly all solid tumors. In this study, we focused on elucidating the effect of canagliflozin (CANA), a new class of antidiabetic agents, on hepatocarcinoma (HCC) tumorigenesis under hypoxia, and demonstrated that CANA could significantly inhibit hypoxia-induced metastasis, angiogenesis, and metabolic reprogramming in HCC. At the molecular level, this was accompanied by a reduction in VEGF expression level, as well as a reduction in the epithelial-to-mesenchymal transition (EMT)-related proteins and glycolysis-related proteins. Next, we focused our study particularly on the modulation of HIF-1α by CANA, which revealed that CANA decreased HIF-1α protein level by inhibiting its synthesis without affecting its proteasomal degradation. Furthermore, the AKT/mTOR pathway, which plays an important role in HIF-1α transcription and translation, was also inhibited by CANA. Thus, it can be concluded that CANA decreased metastasis, angiogenesis, and metabolic reprogramming in HCC by inhibiting HIF-1α protein accumulation, probably by targeting the AKT/mTOR pathway. Based on our results, we propose that CANA should be evaluated as a new treatment modality for liver cancer.

## 1. Introduction

Hypoxia is a major and common feature of clinically relevant solid malignant tumors, including hepatocellular carcinoma (HCC), which is the fifth most common neoplasia with the third highest mortality worldwide [1,2]. Hypoxia, a typical tumor microenvironment, plays an important role in oncogenesis promotion that stimulates metastasis, tumor angiogenesis and glycolysis [3]. Hypoxia-inducible factor 1 (HIF-1) is claimed to be a principal promoter of the tumor hypoxia adaptive response [4]. The biological activity of HIF-1 is determined by HIF-1α expression, and HIF-1α levels are increased due to inactivation of prolyl hydroxylase under hypoxic conditions [5]. As the main regulator of the hypoxic transcriptional response, HIF-1α and its regulated target genes have been identified to be involved in systemic physiological responses to hypoxia, including glycolysis, metastasis and angiogenesis in HCC [6]. HCC is largely dependent on angiogenesis, which can provide a stable energy flow for metastasis, while glycolysis also provides abnormal energy supply for malignant cells in HCC [7]. Because HIF-1 not only regulates oxygen consumption (glycolysis), but also oxygen delivery (angiogenesis) in the hypoxic tumor microenvironment, it is regarded as a promising target in the ongoing search for agents that will inhibit angiogenesis and glycolysis in major solid tumors, including HCC [8,9].

Canagliflozin (CANA) is a kind of sodium-glucose cotransporter 2 (SGLT2) inhibitor which serves as a new hypoglycemic drug that decreases glucose reabsorption in the kidneys [10]. More and more studies indicate that CANA exhibit additional activities independent of the hypoglycemic effect [11,12]. Particularly, recent evidence has highlighted the inhibitory effects of several SGLT2 inhibitors on tumors. Xu et al. [13] demonstrated that CANA could reduce cell viability and increase apoptotic rate in pancreatic cancer. Dapagliflozin, another SGLT2 inhibitor, could inhibit tumor growth and cancer proliferation by targeting the adenosine 5‘-monophosphate (AMP)-activated protein kinase (AMPK)/mammalian target of rapamycin (mTOR) pathway [14]. Besides, the clinical investigation of dapagliflozin among patients with prostate or breast cancer and the clinical investigation of CANA in combination with serabelisib with advanced solid tumors have already been registered [15,16,17]. In addition, we have already demonstrated that CANA could inhibit the growth of HepG2 xenograft tumors in mice and increase the sensitivity of doxorubicin in our previous work [18]. However, the effect of CANA on cancer cells under hypoxia has not been reported and its regulation is yet to be elucidated.

Here, we report anti-tumorigenesis traits of CANA under hypoxia, against human liver cancer cells. First, we have demonstrated the effect of CANA on hypoxia-induced metastasis, angiogenesis and glycolysis. Then the underlying molecular mechanism of CANA was investigated in association with decreasing HIF-1α protein synthesis via the AKT/mTOR pathway.

## 2. Results

### 2.1. CANA Inhibited the Metastasis of HepG2 Cells 

The molecular structure of CANA was shown in Figure 1A. To mimic the tumor microenvironment’s hypoxic condition, we used CoCl_2_ to induce chemical hypoxia in vitro [19]. Firstly, we measured the inhibitory effects different concentrations of CANA on the viability of HepG2 cells under hypoxic conditions for 24 h. Results showed that a drug concentration greater than 50 μM could significantly inhibit the growth of HepG2 cells, while 0–20 μM of CANA did not significantly decrease cellular viability after 24 h of incubation respectively (Figure 1B). Thus, we chose the concentration of 10 and 20 μM of CANA for further investigation in order to exclude the CANA’s cytotoxicity.

To evaluate the role of CANA against the migration capacity of HepG2 cells, wound healing assays and Transwell assays were performed. As shown in Figure 1C,D, in HepG2 cells, CANA treatment significantly reduced the number of migratory cells and decreased the rate of healing in a concentration-dependent manner under CoCl_2_-induced hypoxic conditions. We found that CANA had similar inhibitory effects in Hep3B (Appendix A) and HCCLM3 (Appendix A) cell lines.

### 2.2. CANA Inhibited the Expression of Epithelial-to-Mesenchymal Transition (EMT)-Related Proteins

EMT, a biological process by which epithelial cells acquire invasive mesenchymal stem cell properties and lose polarity, is considered as an important role in promoting metastasis in hepatocellular carcinoma [20]. To further confirm the effects of CANA on EMT, we measured the expression of typical epithelial and mesenchymal markers by Western blotting (Figure 2A,B). Treatment of HepG2 cells with CoCl_2_ led to increased expression of mesenchymal markers (FN1 and snail) and decreased expression of epithelial markers (E-cadherin and ZO-1). The data demonstrated that CANA were remarkably concentration-dependent, decreasing the expression of FN1 and snail and increasing the expression of E-cadherin and ZO-1 compared with the CoCl_2_-treated group. Moreover, as shown in Figure 2C, the administration of CANA reversed the increased β-catenin expression and decreased talin expression, which contributed to induction of the EMT in tumor cells. As ROS is also the inducer of the EMT [21], we then evaluated the intracellular ROS generation by DCFH-DA assay. As shown in Figure 2D, the ROS level in the CoCl_2_ group was higher than that in the control group, and CANA treatment remarkably inhibited ROS expression. These results thus indicated that CANA inhibited the EMT in HepG2 cells.

### 2.3. CANA Inhibited Angiogenesis

Since hypoxia is a key signal for the induction of tumor angiogenesis [22], we determined whether CANA could decrease angiogenesis under hypoxic conditions. In a tube formation assay, the capillary tube formation of HUVECs cells was markedly increased under hypoxic conditions, while CANA had a concentration-dependent inhibitory effect on the tube formation of HUVEC cells (Figure 3A). We found that CANA showed similar inhibitory effects in Hep3B and HCCLM3 cell lines (Appendix A). We also performed the vasculogenic mimicry assay, and found that CANA also inhibited vascular mimicry (VM) formation of HepG2 cells under hypoxia (Figure 3B). Because VEGFA is the master effector of the angiogenic response in cancers [23], we further examined the expression of VEGFA by Western blot and immunohistochemistry (Figure 3C–E). Results showed that CANA significantly suppressed VEGFA protein expressions both in vitro and in vivo.

### 2.4. CANA Inhibited Glycolysis under Hypoxia Condition

Non-functioning vascularization in tumors results in intermittent hypoxia that forces cells to rely on glycolysis [23]. Treatment with CoCl_2_ induced an obvious increase of ATP level, glucose uptake and lactate production HepG2 cells, while administration of CANA greatly weakened these events in a concentration-dependent dose (Figure 4A). In addition, we determined the protein expression level of HK2, PFKFB3, LDHA and GLUT1, which are several key enzymes involved in the regulation of glycolytic pathways, in cells by using Western blotting; the data showed that HK2, PFKFB3, LDHA and GLUT1 expression was increased under hypoxic conditions (Figure 4B–E). However, their expressions were remarkably decreased after administration of CANA, which may imply that CANA could inhibit the glycolysis in HepG2 cells.

### 2.5. CANA Inhibited Accumulation of HIF-1α

Swiss Target Prediction was used to predict the targets of CANA. For liver cancer, we retrieved 7817 genes (DATA S1) from the Gene Cards database. We mapped the putative targets of CANA (DATA S2) to the related targets of liver cancer by drawing the Venn diagram and 55 overlapping targets were obtained (Figure 5A). To find the key regulators for overlapping genes based on transcription factors, we submitted the candidate targets to the TRRUST2.0 platform [24]. As shown in Figure 5A, HIF-1α is the most potential key regulator (selected based on *p* value). Based on bioinformatics analysis, we selected the HIF-1α for the preliminary investigation. Results obtained from Western blotting indicated that CANA decreased CoCl_2_-induced HIF-1α protein expression in vitro significantly in a concentration-dependent manner (Figure 5B and Appendix A). Meanwhile, we also tested the effect of LW-6(HIF-1α inhibitor) and we could find the synergistic inhibitory effect of CANA and LW-6 on HIF-1α protein expression in vitro (Figure 5C). We next evaluated the effect of CANA administration on HIF-1α expression levels in vivo in a HepG2 xenograft mice model. HIF-1α protein expression levels were decreased upon CANA administration as evident from immunohistochemistry (Figure 5C). Those results strongly suggested that CANA inhibited HIF-1α expression both in vivo and in vitro.

### 2.6. CANA Triggerd HIF-1α Reduction through the AKT/mTOR Pathway through Inhibiting HIF-1α Protein Synthesis

To understand the molecular mechanism by which CANA suppresses HIF-1α, we initially investigated whether CANA could affect HIF-1α at transcriptional level under the hypoxia condition. RT-PCR analysis revealed that HIF-1α mRNA levels almost remained unchanged by treatment with CANA in hypoxia (Figure 6A). Then, we examined the HIF-1α post transcriptional regulation. HIF-1α is degraded via prolyl hydroxylases (PHDs)-mediated hydroxylation and von–Hippel Lindau protein (pVHL)-mediated polyubiquitination [25]. Since the hydroxylation does not occur under hypoxic conditions, we used MG132, a proteasome inhibitor, to test whether inhibition of HIF-1α protein by CANA was due to enhanced proteasomal degradation of the protein (Figure 6B). However, MG132 was unable to inhibit CANA’s effect on the decrease of HIF-1α expression, which indicated that CA possibly interfered with HIF-1α protein synthesis (Figure 6C). In addition, we treated cells with cycloheximide (CHX), a new protein synthesis inhibitor, to test stability of HIF-1α (Figure 6D). The result showed that HIF-1α was not degraded faster than in control cells, which also indicated that CANA did not modulate proteasomal degradation of HIF-1α. 

In addition, the overlapping targets we got before were submitted to the Metascape platform to establish the protein–protein interaction network. We used MCODE to obtain a potential protein functional module with the best score for further WikiPathway enrichment analysis. As shown in Figure 6E, results implied that the PI3K/AKT/mTOR signaling pathway might be the key pathway. It is also well-established that the AKT/mTOR axis have an important role in the up-regulation of HIF-1α translation [26]. Thus, we measured some key proteins, including AKT, *p*-AKT, mTOR, *p*-mTOR, P70S6K, and *p*-P70S6K under chemically-induced hypoxia. Results indicated that administration of CANA remarkably decreased the expression levels of *p*-AKT, *p*-mTOR and *p*-P70S6K in HepG2 cells in a concentration-dependent manner (Figure 6F,G). To confirm the AKT/mTOR signaling pathway was involved in the regulation of HIF-1α protein expression, we treated HepG2 cells in combination with LY294002 (*p*-AKT inhibitor) and assessed the levels of HIF-1α expression. Treatment of HepG2 cells with LY294002 alone, or in combination with CANA, inhibited HIF-1α expression in cells (Figure 6H). Then, we treated HepG2 cells in combination with insulin, a known activator of *p*-AKT. Results showed that CANA could inhibit insulin-induced HIF-1α and AKT phosphorylation (Figure 6I). These results indicated that CANA could trigger HIF-1α reduction through the AKT/mTOR pathway through inhibiting HIF-1α protein synthesis.

## 3. Discussion

Tumor hypoxia is a major limitation of current tumor therapy, which is of clinical relevance, and the hypoxic microenvironment of solid tumors promotes tumorigenesis [3]. Tumor cells existing in a hypoxic environment adapt the downstream processes of hypoxia to eventually adopt a more aggressive phenotype [27]. This study was the first to investigate the effect of CANA on metastasis, EMT, angiogenesis and glycolysis under hypoxia in liver cancer and explore the potential molecular mechanism.

Previous studies have already proved that hypoxic tumor cells are more aggressive, with a better ability to metastasize [28]. Therefore, we investigated the effect of CANA on tumor cell migration in HCC. We demonstrated that CANA remarkably inhibited the tumor cell migration brought about by hypoxia. Mechanistically, hypoxia was shown to influence metastasis behavior of tumor cells via EMT, which happens in cancerogenesis in many solid tumors and involves epithelial traits loss and mesenchymal characteristic acquisition to acquire plastic and mobile abilities [29,30]. Hypoxia-induced EMT is marked by a decrease in epithelial-associated gene expression and an increase in mesenchymal-like gene expression [31]. In this study, for the first time, CANA was found to significantly reduce the EMT process, which suggest that the decreased expression of E-cadherin, ZO-1 and the increased expression of FN-1, snail were alleviated by CANA. In addition, compelling evidence highlights ROS as crucial conspirators in EMT engagement [32]. As expected, CANA decreased hypoxia-induced overproduction of ROS.

Angiogenesis is a crucial process for tumor growth, and metastasis for tumors are dependent on the blood vessels to acquire nutrients [33]. The hostile hypoxic environment forces tumors to intravasate, circulate, and relocate to new and unaffected tissues by permeable and heterogeneous vascular system [34]. In our study, we demonstrated the effect of CANA in inhibiting angiogenesis of HCC under hypoxia. VEGF, a key regulator of angiogenesis, was also inhibited by CANA administration in vivo and in vitro. Vasculogenic mimicry, which has been found in many malignant tumors including HCC, results in the formation of a new blood supply system in aggressive tumor cells [35]. Results showed that HepG2 cells gain the VM forming ability under hypoxia and CANA for the first time, thus proving to inhibit the mimicry formation of HepG2 cells. The development of targeted drugs for angiogenesis has always been a hot topic in tumor therapy, while the clinical effects of these drugs have not been satisfactory. Numerous studies suggested it may be related to the activation of VM [36,37]. Therefore, CANA may be a hopeful potential candidate drug for VM; although, it is undeniable that more in vitro and in vivo evidence is needed. 

Tumorigenesis is considered not only an inherent process of genetic instability, but also the result of metabolic reprogramming which is characterized by an elevated rate of glycolysis [38]. A major consequence of an elevated rate of glycolysis is that glucose carbon is converted primarily to lactate which causes acidification of the tumor microenvironment, and the decrease in extracellular pH can promote tumor cell migration [39]. Under hypoxia, glucose consumption, lactate production and ATP production were enhanced in HepG2 cells, indicating that glycolysis was triggered; meanwhile, most glycolytic enzymes including GLUT-1, HK2, PFKFB3 and LDHA are HIF-1α targets that are transcriptionally activated in response to hypoxia. However, CANA administration could reduce glycolysis to reprogram HCC cell metabolism under hypoxia.

HIF-1α, the principal regulator of hypoxia, is capable of modifying 1–2% of the genome [40] to adapt to a decrease in partial pressure of oxygen and nutrient availability, and HIF-1α-induced genes are those mediated a large number of effects including increased glycolysis, angiogenesis, EMT, and resistance to the tumor chemotherapy [41]. Through bioinformatic analysis, we thought CANA probably inhibited tumorigenesis by targeting HIF-1α in HCC. Therefore, after proving the anti-metastasis, anti-angiogenesis and anti-glycolysis potential of CANA, we next focused our attention on the modulation of HIF-1α by CANA. We have shown here that the accumulation of HIF-1α protein under hypoxia was significantly reduced in HCC cells and in the HepG2 xenograft mice model. We also demonstrated the synergistic inhibitory effect of CANA and LW-6(HIF-1α inhibitor) on cell migration and angiogenesis (Appendix A). Further studies on the mechanisms of HIF-1α regulation by CANA revealed that CANA inhibited HIF-1α accumulation without affecting its proteasomal degradation or mRNA levels. Alternatively, CANA probably affect the de novo synthesis of HIF-1α.

It is well-established that major hypoxia-associated signaling pathways including PI3k/AKT/mTOR, mitogen-activated protein kinases (MAPK) and nuclear factor kappa-B (NFκB) have important roles in the regulation of HIF-1α expression [4,42,43]. Verified by bioinformatics analysis and experiment, CANA treatment can reduce expressions of *p*-AKT, *p*-mTOR and *p*-P70S6K levels in vitro during hypoxic conditions, which might cause a reduction in HIF-1α translation. For further validation of the involvement of the AKT/mTOR axis in CANA-induced inhibition of HIF-1α, we evaluated the effect of CANA in the presence or absence of the AKT inhibitor (LY294002) and inducer (insulin). Results marked that CANA probably acts through the AKT/mTOR pathway in the reduction of HIF-1α expression. 

In our previous work, we have demonstrated that CANA inhibited tumor growth and anticancer resistance [18]. We further explored the effect of CANA on liver cancer. Here, we firstly demonstrated that the inhibitory effect of CANA on metastasis, EMT, angiogenesis and glycolysis under hypoxia is largely achieved through the reduction of HIF-1α synthesis via the AKT/mTOR pathway. Repurposing CANA is pioneering research into the treatment of cancers, and this study provides a new insight into the molecular mechanism of the anticancer activity of CANA, which indicates that SGLT2i could be considered as an optional therapy for HCC.

## 4. Materials and Methods

### 4.1. Chemicals, Reagents and Antibodies

CANA, MG132, LW-6 and cycloheximide (CHX) were obtained from MedChemExpress (Monmouth Junction, NJ, USA). Cobaltous chloride (CoCl_2_) was obtained from Sigma-Aldrich (Chicago, NJ, USA). The specific primary antibodies used for Western blotting analysis against protein kinase B (AKT) (1:1000), *p*-AKT (1:1000), mTOR (1:1000), *p*-mTOR (1:1000), HIF-1α (1:1000), fibronectin (FN-1) (1:1000), lactate dehydrogenase A (LDHA) (1:1000), glucosetransporter1 (GLUT1) (1:1000), hexokinase 2 (HK2) (1:1000), 6-phosphofructo-2-kinase/fructose-2,6-biphosphatase 3 (PFKFB3) (1:1000), β-catenin (1:1000), E-cadherin, snail, zonula occludens-1 (ZO-1) (1:1000) and talin (1:1000) were purchased from Cell Signaling Technology (Boston, MA, USA), while actin (1:50,000) was purchased from Sigma-Aldrich^®^ (Darmstadt, Germany), and vascular endothelial growth factor A (VEGFA) (1:500), P70S6K (1:2000), and *p*-P70S6K (1:2000) were purchased from Abclonal (Wuhan, China). The secondary antibodies were purchased from Cell Signaling Technology (Boston, MA, USA).

### 4.2. Cell Culture

Human umbilical vein endothelial cells (HUVECs), HepG2, Hep3B and HCCLM3 cells were purchased from the Cell Resource Centre of the Shanghai Institute for Biological Science, Chinese Academy of Sciences, Shanghai, China. The cells were maintained at 37 °C in a humidified atmosphere of 95% air and 5% CO_2_. The cell culture medium consisted of Dulbecco’s modified Eagle medium (DMEM, Gibco, Waltham, MA, USA) with high glucose and Ham’s F12K medium (Procell Life Science&Technology, Wuhan, China) supplemented with 10% fetal bovine serum (FBS; Invitrogen Carlsbad, Carlsbad, CA, USA) and 1% penicillin-streptomycin antibiotic (Gibco™, ThermoFisher Scientific, Waltham, MA, USA). CoCl_2_ (Sigma-Aldrich ^®^, Darmstadt, Germany) was used to model chemical hypoxia.

### 4.3. Cytotoxicity Assay

Briefly, HepG2 cells were seeded in 96-well plates at a density of 1 × 10^4^ cells/well and then incubated with 100 μM CoCl_2_ and different concentrations of CANA for 24 h. Next, 10 μL of thiazolyl blue tetrazolium bromide (MTT) working solution (5 mg/mL) was added to each well and the cells were further cultivated for 4 h. Afterward, the formazan crystals were dissolved in 200 μL of dimethyl sulfoxide (Sigma-Aldrich^®^, Darmstadt, Germany) and the absorbance was quantitatively assayed on an Epoch microplate spectrophotometer (wavelength: 490 nm; Bio-Tek, Winooski, VT, USA)

### 4.4. Anticancer Metastasis Activity Assay

#### 4.4.1. Wound Healing Assay

HepG2 cells were seeded in 96-well plates and grown to form a monolayer. Straight wounds were created in the cell monolayers with a 200-μL sterile pipette tip and detached cells were washed twice using phosphate-buffered saline. Subsequently cells were exposed to 100 μM CoCl_2_ and different concentrations of CANA for 24 h. Images were captured at 100× magnification at 0 and 24 h. The cell migration rate was calculated as % = (wound width at 0 h–wound width at 24 h)/wound width at 24 h%.

#### 4.4.2. Transwell Assay

The migrative ability of HepG2 cells was assessed using Transwell chambers with a pore of 8 μm (Corning Costar, New York, NY, USA). Briefly, the cells were seeded onto the upper Transwell chamber at a density of 5 × 10^4^ cells/well with serum-free DMEM. The lower Transwell chamber contained DMEM with 10% FBS. Following 24 h migration at 37 °C, the migrated cells were fixed in 4% paraformaldehyde for 20 min, and stained with crystal violet for 10 min. Three visual fields were randomly selected from each Transwell filter and captured at 100× magnification under a Leica DMI6000 B inverted microscope (Leica Microsystems, Weztlar, Germany). The average number of cells that migrated through the Transwell filter was calculated using ImageJ software (National Institutes of Health, Bethesda, Maryland, USA).

### 4.5. Reactive Oxygen Species (ROS) Assay

The ROS-specific fluorescence probes 5- (and 6-) chloromethyl-2′,7′-dichlorodihydrofluorescein diacetate, acetyl ester (DCF-DA) (Beyotime, Shanghai, China) was utilized to determine the intracellular ROS level. Images was photographed under a Leica DMI6000 B inverted microscope (Leica Microsystems, Weztlar, Germany), and the excitation/emission wavelength of 488/525 nm was examined using a microplate reader (Tecan, Männedorf, Switzerland). The cell was variability tested by MTT. Relative ROS level was analyzed by normalizing to the control group.

### 4.6. Tube Formation Assay

HepG2 cells were exposed to 100 μM CoCl_2_ and different concentrations (10 μM and 20 μM) of CANA for 24 h to obtain the condition medium. Fifty microliters of Matrigel (Becton, Dickinson and Company, Franklin Lakes, NJ, USA) were placed in each well of the 96-well plates and incubated at 37 ℃ for 40 min. HUVEC cells (4 × 10^4^) were seeded and incubated with the condition medium, or 5 × 10^4^ HepG2 cells were seeded. After 12 h of treatment, the tube formation was photographed with a Leica DMI6000 B inverted microscope (Leica Microsystems, Weztlar, Germany), and the vessel areas were calculated using ImageJ software.

### 4.7. Cellular Glucose, Lactate and ATP Analysis

Cellular ATP level was measured using quantification kit (Beyotime Biotechnology, Shanghai, China) according to the manufacturer’s instructions. The Enhanced Bradford protein assay kit (Beyotime Biotechnology, Shanghai, China) was used to quantify protein concentration. Extracellular glucose and lactate levels (Elabscience Biotechnology, Wuhan, China) were measured using quantification kits, respectively.

### 4.8. Western Blot Assay

Briefly, total proteins were extracted and the Enhanced Bradford protein assay kit was used to quantify protein concentration. Equal amounts of protein (30 μg/lane) from each sample were resolved by 10% or 12.5% sodium dodecyl sulfate polyacrylamide gel (Epizyme Biotech, Shanghai, China) electrophoresis (180 V, 50 min), followed by transfer (200 mA,120 min) to nitrocellulose transfer membranes (Pall, New York, NY, USA). Membranes were blocked with 5% (g/mL) non-fat dry milk (Anchor, New Zealand) for 2 h and probed with primary antibodies at 4 °C overnight, followed by incubation with the respective second antibody for 1 h at room temperature. The immunoreactivity was visualized by enhanced chemiluminescence solution (Thermo Fisher Scientific, Waltham, MA, USA).

### 4.9. Quantitative Real-Time PCR Analysis

RNA was extracted using RNAiso Plus (TaKaRa Biotechnology, Dalian, China) and measured using a NanoDrop 2000 spectrophotometer (Thermo Fisher Scientific, Waltham, MA, USA). Total RNA (500 ng) in each sample was reverse-transcribed to cDNA in 10 µL reactions using an Evo M-MLV RT Premix Kit (Accurate Biology, Changsha, China) following the manufacturer protocol. The expression levels of mRNAs were assayed via quantitative RT-PCR on a qTOWER 2.0 Real Time PCR System with two channels (Analytik Jena AG, Jena, Germany) using a SYBR Green qPCR master mix (Accurate Biology, Changsha, China). All data were normalized to the housekeeping gene *β-actin*. Primers (*human β-actin*: CATGTACGTTGCTATCCAGGC (Forward), CTCCTTAATGTCACGCACGAT (Reverse); HIF-1α: GAACGTCGAAAAGAAAAGTCTCG (Forward), CCTTATCAAGATGCGAACTCACA (Reverse)) were purchased from GENEWIZ (Cambridge, MA, USA).

### 4.10. Animal Experiment

Animal experiment was performed according to the previous method [18]. The protocol was approved by the Life Ethics Committee of Tsinghua Shenzhen International Graduate School, Tsinghua University, China. In brief, four-week-old male BALB/c Slc-nu/nu mice were obtained from Guangzhou Medical Animal Centre (Guangzhou, China). After a one-week adaptation period, 2.0 × 10^6^ HepG2 cells were suspended in a 200 μL mixture of DMEM medium and Matrigel (*v/v*, 9:1; Becton, Dickinson and Company, FL, NJ, USA) that was injected into the bilateral flanks of mice. The nude mice that successfully modeled were divided into a normal-control group and a CANA-treated group. The normal-control group was treated with 0.5% carboxymethylcellulose (g/mL) by gavage, while the CANA-treated group was treated with CANA at a dosage of 50 mg/kg by gavage once a day [18]. All mice were sacrificed after four weeks of treatment and the tumor tissue were obtained for further analysis.

### 4.11. Immunohistochemical Staining

In brief, tumor tissue slices were placed in xylene three times (15 min/time). Then, the tissue slices were placed in 100, 85, and 75% ethanol at each stage for 5 min. Afterward, tissue slices were placed in 3% hydrogen periodate at room temperature in darkness for 25 min, and then rinsed with PBS three times (5 min/time). Then, the primary antibody VEGFA (Abclonal, Wuhan, China) and HIF-1α (Cell Signaling Technology, Boston, MA, USA) were incubated overnight at 4 °C separately, and the secondary antibody (HRP labeled, Cell Signaling Technology, Boston, MA, USA) was incubated for 50 min at 37 °C. Diaminobenzidine was used to visualize the slices. The slices were dehydrated and sealed with neutral balsam and observed under a microscope (Leica Microsystems, Weztlar, Germany).

### 4.12. Pathway Enrichment Analysis and Transcriptional Regulatory Analysis

The possible targets of CANA were obtained from Swiss Target Prediction [44] (http://www.swisstargetprediction.ch/, accessed on 21 June 2021), a web server for small molecule drug target prediction. Information on liver cancer-associated target genes was collected from GeneCards [45] (http://www.genecards.org/, accessed on 21 June 2021), a database containing details of disease-related genes and proteins. Metascape [46] (https://metascape.org, accessed on 22 June 2021) was used to further analyse the pathway enrichment. TRRUST [24] (https://www.grnpedia.org/trrust/ accessed on 25 June 2021) is a useful prediction tool for human and mouse transcriptional regulatory networks. A bubble plot was created by http://www.bioinformatics.com.cn (accessed on 29 June 2021), a free online platform for data analysis and visualization.

### 4.13. Statistical Analysis

GraphPad Prism 8.0 software was used for the statistical analysis. All the data were repeated three or more times independently and expressed as mean ± standard deviation (S.D.). Differences with statistical significance between groups were calculated by ANOVA followed by Tukey’s post hoc test. *p* < 0.05 was considered statistically significant.

## Figures and Tables

**Figure 1 ijms-22-13336-f001:**
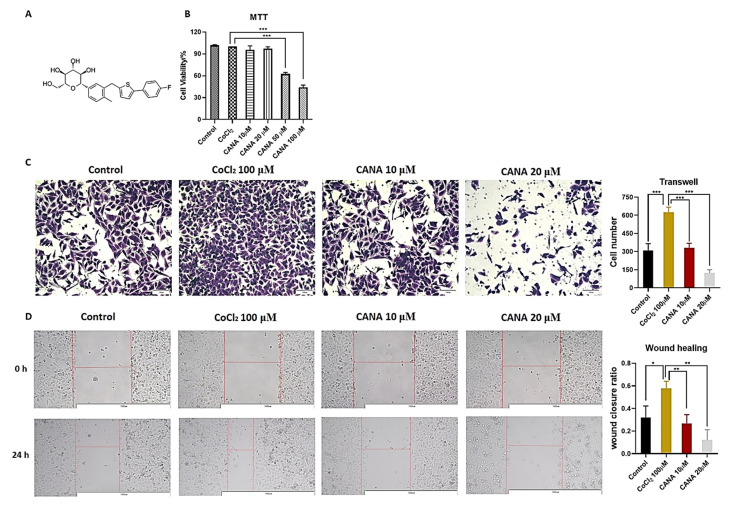
CANA inhibited hypoxia-induced metastasis. (**A**) Chemical structure of CANA. (**B**) Cytotoxic effects after chemical hypoxia (CoCl_2_ 100 μM) and CANA (0–100 μM) intervention for 24 h were evaluated by MTT assay. Transwell (200X) (**C**) and wound healing assays (100X) (**D**) were performed to determine the migration of HepG2 cells. Data were shown as means ± S.D. (*n* = 3); * *p* < 0.05, ** *p* < 0.01, and *** *p* < 0.001. Control, normal untreated control; CANA, CANA-treated CoCl_2_ (100 μM)-induced group.

**Figure 2 ijms-22-13336-f002:**
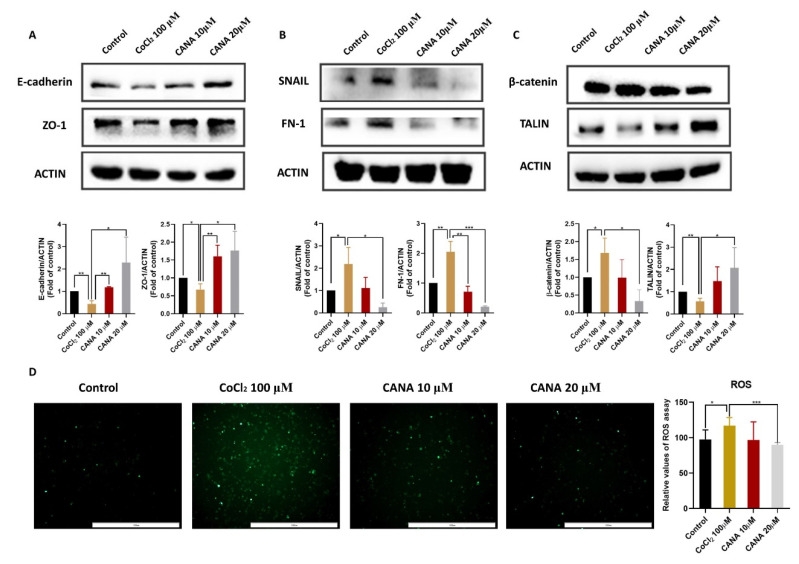
CANA inhibited hypoxia-induced epithelia-to-mesenchymal transition. (**A**) Effect of CANA on the protein expression of the epithelial marker in hypoxia-induced HepG2 cells after 24 h. (**B**) Effect of CANA on the protein expression of the mescenchymal marker in hypoxia-induced HepG2 cells after 24 h. (**C**) Effect of CANA on the protein expression of β-catenin and talin in hypoxia-induced HepG2 cells after 24 h. (**D**) Effects of CANA on reactive oxygen species (ROS) in hypoxia-induced HepG2 cells after 24 h (100×). Data were shown as means ± S.D. (*n* = 3); * *p* < 0.05, ** *p* < 0.01, and *** *p* < 0.001. Control, normal untreated control; CANA, CANA-treated CoCl2 (100 μM)-induced group.

**Figure 3 ijms-22-13336-f003:**
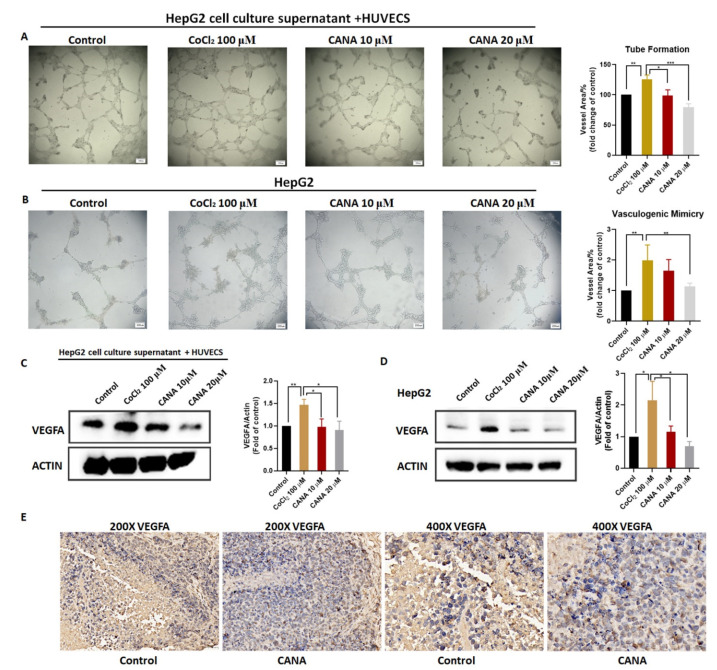
CANA inhibited hypoxia-induced angiogenesis. HepG2 cells were cultured in DMEM medium or CoCl_2_ 100 μM and CANA (10 or 20 μM) for 24 h to obtain the condition medium. (**A**) Tube formation was performed with condition medium to determine the angiogenic ability of HUVECs (50X). (**B**) Tube-like structure formation on Matrigel in HepG2 cells (50X). Protein expression levels in (**C**) HUVEC cells with condition medium or in (**D**) HepG2 cells. (**E**) Immunohistochemical staining for the VEGFA protein in tumor specimens from xenografts (200X, 400X). Data were shown as means ± S.D. (*n* = 3); * *p* < 0.05, ** *p* < 0.01, and *** *p* < 0.001. Control, normal untreated control; CANA, CANA-treated CoCl_2_ (100 μM)-induced group.

**Figure 4 ijms-22-13336-f004:**
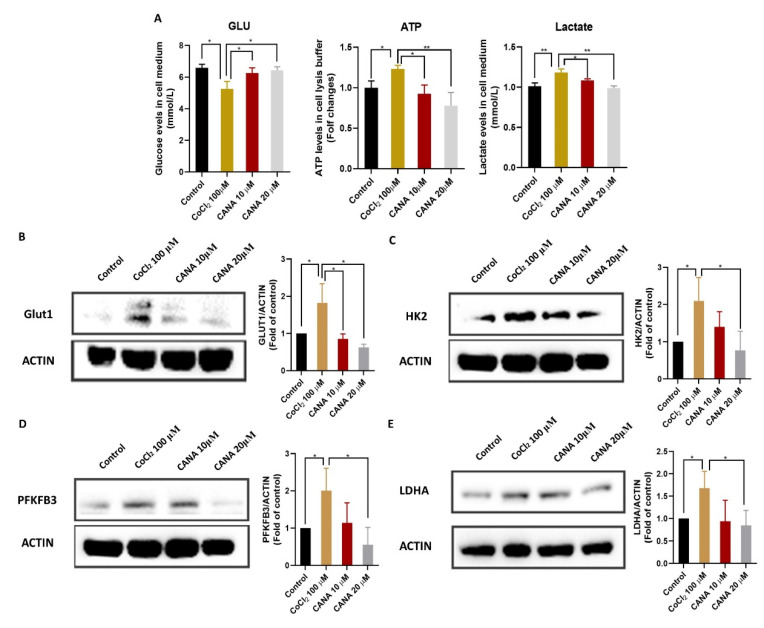
CANA inhibited hypoxia-induced glycolysis. (**A**) Effect of CANA or/and chemical hypoxia on extracellular glucose levels, extracellular lactate levels and extracellular ATP levels in HepG2 cells. (**B**–**E**) Effect of CANA or/and chemical hypoxia on protein expression of glycolytic enzymes in HepG2 cells. Data were shown as means ± S.D. (*n* = 3); * *p* < 0.05 and ** *p* < 0.01. Control, normal untreated control; CANA, CANA-treated CoCl_2_ (100 μM)-induced group.

**Figure 5 ijms-22-13336-f005:**
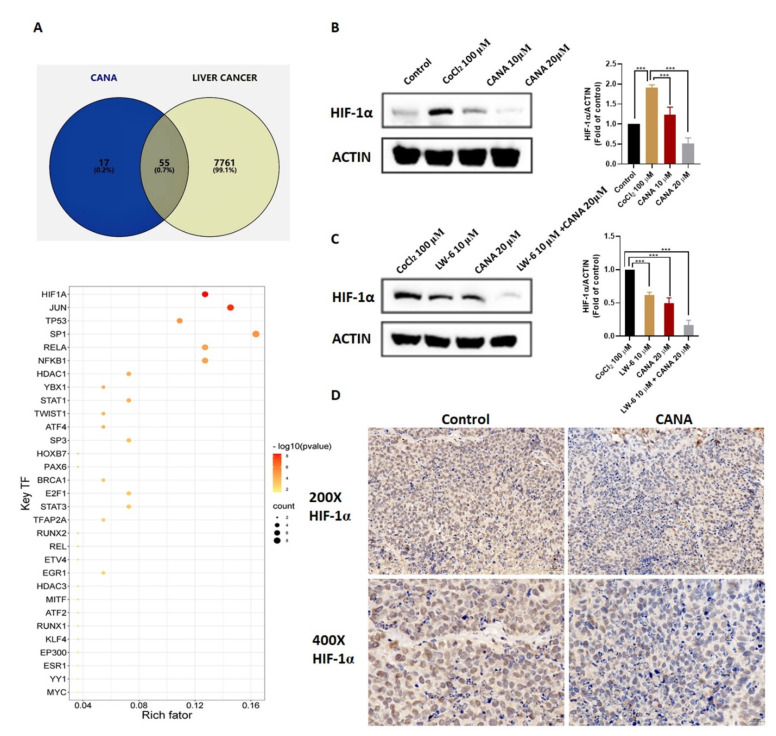
CANA inhibited accumulation of HIF-1α. (**A**) Venn diagram of the common targets between liver cancer-related targets and CANA potential targets and bubble plot of the key regulators for overlapping genes. (**B**) Expression of HIF-1α protein in HepG2 cells exposed to CoCl_2_ in the presence or absence of CANA. (**C**) Expression of HIF-1α protein in HepG2 cells exposed to CoCl_2_ in the presence of LW-6, CANA, or both. (**D**) Representative tumor tissue sections showing expression of HIF-1α protein detected immunohistochemically in the indicated groups (200X, 400X). Data were shown as means ± S.D. (*n* = 3); *** *p* < 0.001. Control, normal untreated control; CANA, CANA-treated CoCl_2_ (100 μM)-induced group; LW-6, LW-6-treated CoCl_2_ (100 μM)-induced group; LW-6 + CANA, LW-6-treated CANA-treated CoCl_2_ (100 μM)-induced group.

**Figure 6 ijms-22-13336-f006:**
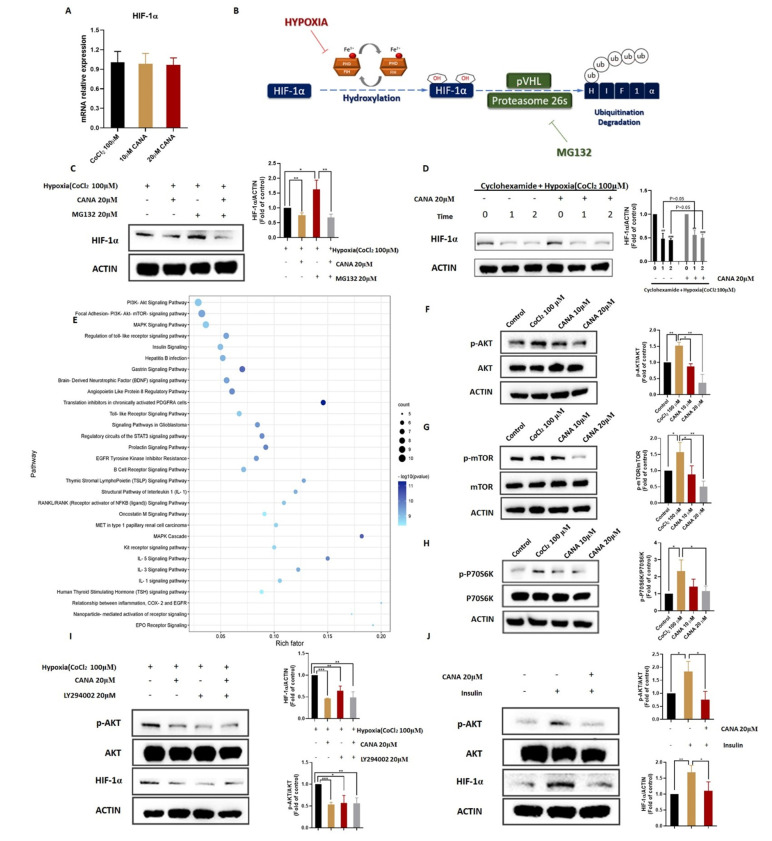
CANA triggered HIF-1α reduction through the AKT/mTOR pathway through inhibiting HIF-1α protein synthesis. (**A**) Effect of CANA or/and chemical hypoxia on mRNA expression of HIF-1α in HepG2 cells. (**B**) Schematic summary of HIF-1α hydroxylation and its proteasome-dependent degradation pathway. (**C**) HIF-1α protein expression in HepG2 cells treated with CANA or proteasome inhibitor MG132 under chemical hypoxia. (**D**) HIF-1α protein expression in HepG2 cells treated CHX and with/without CANA under chemical hypoxia. (**E**) WikiPathway enrichment analysis of the common targets. (**F**–**H**) Effect of CANA or/and chemical hypoxia on protein expression of AKT/mTOR pathway-related molecules in HepG2 cells. (**I**) HepG2 cells were treated with LY294002 or/and CANA under hypoxia followed by Western blots analysis of HIF-1α, AKT, and *p*-AKT. (**J**) HepG2 cells were treated with insulin or/and CANA followed by Western blots analysis of HIF-1α, AKT, and *p*-AKT. Data were shown as means ± S.D. (*n* = 3); * *p* < 0.05, ** *p* < 0.01, *** *p* < 0.001. Control, normal untreated control; CANA, CANA-treated CoCl_2_ (100 μM)-induced group.

## Data Availability

The data presented in this study are available on request from the corresponding author.

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
