# Peer review of "Canagliflozin Modulates Hypoxia-Induced Metastasis, Angiogenesis and Glycolysis by Decreasing HIF-1α Protein Synthesis via AKT/mTOR Pathway"

_ijms, 2021, doi:10.3390/ijms222413336_

Round 1
Reviewer 1 Report
Unfortunately, this manuscript is not subject to peer review and should be rejected, since it has a number of fundamental shortcomings.
The fact that CANADA can inhibit cancer hepatocellular cells is not novel: DOI: 10.1371/journal.pone.0232283, DOI: 10.1038/s41419-019-1646-6, DOI: 10.1038/s41598-018-19658-7, for example.The evidence base for the presented molecular mechanism is highly questionable.
Only one HCC cell line was used in the studies. Authors should have added at least one more HСС line like Huh7, for example.
Evidence for an effect on angiogenesis is completely inadequate, staining and analysis of the number and size of vessels in xenografts are lacking.
There is no direct evidence of the effect of CANA on HIF and ACT/mTOR signaling. shRNA/CRISPR for HIF and analysis of p-S6 and p-AMPK (conformable to hypoxic condition) are absence.
A direct experimental evidence of the CANA-induced effect on glycolysis and metastasis, which is stated in the title of the article, is also completely lacking.
Without these studies, the authors' conclusions are inappropriate.
Author Response
Following are our point-to point response to the comments of reviewers.
Review Report 1
The fact that CANADA can inhibit cancer hepatocellular cells is not novel: DOI: 10.1371/journal.pone.0232283, DOI: 10.1038/s41419-019-1646-6, DOI: 10.1038/s41598-018-19658-7, for example.The evidence base for the presented molecular mechanism is highly questionable.
Thank you for your useful information. About the studies: DOI: 10.1371/journal.pone.0232283 and DOI: 10.1038/s41419-019-1646-6, we have checked these papers. These studies were focused on the effects of CANA on proliferation and targeting metabolic reprogram.
For another study: DOI: 10.1038/s41598-018-19658-7, we have known that this study provides evidences that CANA prevented ectopic fat accumulation in the liver via promoting healthy adipose expansion and inhibited the development of hepatic fibrosis and hepatocellular carcinoma. It mainly focused on NASH-associated hepatocellular carcinoma which was a completely different model from us.
Actually, all three studies mentioned by the reviewer have different pathological model from us. Tumor-induced hypoxia is one of major limitations of current tumor therapy. The hypoxia microenvironment of solid tumors promotes tumorigenesis and causes a series of dysfunction. So, we choose to construct a hypoxia in vitro model. And our study was the first to investigate the effect of CANA on metastasis, EMT, angiogenesis and glycolysis under hypoxia. At least we added some new or additional evidence for its potential treatment in tumors.
Only one HCC cell line was used in the studies. Authors should have added at least one more HСС line like Huh7, for example.
As required, we added new cell lines(Hep3B and HCCLM3) and updated new data in newly supplement materials.
Evidence for an effect on angiogenesis is completely inadequate, staining and analysis of the number and size of vessels in xenografts are lacking.
We admitted this is a limit for our manuscript. Due to the limited time and resource, it was very difficult for us to conducted these animal experiments. We hope we can complete these experiments in a separate study in the future.
There is no direct evidence of the effect of CANA on HIF and ACT/mTOR signaling. shRNA/CRISPR for HIF and analysis of p-S6 and p-AMPK (conformable to hypoxic condition) are absence.
As required, we have added data of p-S6. We have used HIF inhibitor LW6 to inhibit the function of HIF and observed the relevant and necessary effect. It was very difficult for us to conduct the experiments of shRNA/CRISPR in such a short time. About p-AMPK, lots of studies have reported this activity of CANA (DOI: 10.3389/fphys.2018.01575; DOI: 10.1016/j.bcp.2018.03.013; DOI: 10.2337/db16-0058). So, we did not test this activity.
A direct experimental evidence of the CANA-induced effect on glycolysis and metastasis, which is stated in the title of the article, is also completely lacking.
In Figure 1, through would healing and transwell assay, we demon demonstrated that CANA remarkably inhibited the tumor cell migration brought about by hypoxia. In Figure 2, CANA was found to reduce the EMT process, which suggest that the decreased expression of E-cadherin, ZO-1 and the increased expression of FN-1, snail were alleviated by CANA. These results suggest that CANA do have some effects on metastasis.
In Figure 4, we tested the protein expression level of HK2, PFKFB3, LDHA and GLUT1, which are several key enzymes involved in the regulation of glycolytic pathways under hypoxia after administration of CANA. We also tested glucose consumption, lactate and ATP productions which were associated with glycolysis. We believe our experiments in Figure 4 have provided some evidences for the CANA-induced effects on glycolysis.
Without these studies, the authors' conclusions are inappropriate.
As you required, we do our best to conduct some necessary experiments.
Reviewer 2 Report
The paper by Luo et al. showed the role of an antidiabetogenic agent called canagliflozin (CANA) in many tumor events (e.g., metastasis, angiogenesis, and glycolysis) in HCC cells. Authors conluded that CANA decreased metastasis, angiogenesis and metabolic reprogramming in HCC by inhibiting HIF-1α accumulation probably by targeting the AKT/mTOR pathway. They further emphasized that CANA may be considered as a novel treatment modality for liver cancer.
Although interesting, this manuscript needs some improvements. There are many typing errors over the manuscript or lacking words. Paper should be carefully revised to improve its readability.
In introduction, more emphasis on the angiogenesis and aerobic glycolysis should be given in the context of liver cancer.
In methods, authors stated “that CANA was administered to the animals at a dosage of 50 mg/kg by gavage”. Is there a previous protocol for the dose or is it often take by diabetic individuals? This may be better linked.
In results, why do authors defined the use of 10-20 µM CANA and not applied the IC50 which was near to the 50 µM? These doses were unable to induce metabolic changes.
Figure 3 is duplicated from Figure 1. The legend of figure 3 is described as angiogenesis but it appears the results of metastasis. Modify accordingly.
Predicted targets of CANA and liver cancer could be added as supplementary tables for further consultation.
Page 6, “Those results strongly suggested that CANA inhibited HIF-1α expression both in vivo and in vitro”. These results only demonstrated the HIF1 inhibition in vivo but not in vitro. I suggest removing “in vitro” from the text.
The heat map of figure 5 is only related to key regulators based on transcription factors? If do, a better description should be added to the text.
Which software the Figure 6 E was created? It was enriched the terms based on the 55 shared targets? Specify accordingly.
Discussion, page 9. Authors reported that CANA reduces VEGF (line 246). Which subunit ? If VEGFA, these results could be strengthen with other studies demonstrating the role of small molecules in reducing their levels (PMID: 33145353, PMID: 30755242, PMID: 28398226).
Minor comments
Add dilutions to the antibodies used in WB and IHC
Add the concentrations of CANA used to treat HepG2, vehicle, and control group.
In statistical analysis, describe the graphical software that was used.
Page 4, line 103. Change “As shown in Fig. 2C” by “Fig. 2 D”.
Page 4, figure legend. Change “Figure 1” by “Figure 2”.
Inform in methods whether experiments were done in technical and biological triplicate.
